# Network Meta-Analysis: Effect of Cold Stress on the Gene Expression of Swine Adipocytes *ATGL*, *CIDEA*, *UCP2*, and *UCP3*

Zhenhua Guo [1],[*],[†] , Lei Lv [2],[†], Di Liu [1],[*], Hong Ma [1], Liang Wang [1], Bo Fu [1] and Fang Wang [1]

[1]  Key Laboratory of Combining Farming and Animal Husbandry, Institute of Animal Husbandry, Heilongjiang Academy of Agricultural Sciences, Ministry of Agriculture and Rural Affairs, No. 368 Xuefu Road, Harbin 150086, China

[2]  Wood Science Research Institute of Heilongjiang Academy of Forestry, No. 134 Haping Road, Harbin 150080, China

[*]  Correspondence: guozhenhua@haas.cn or gzhh00@163.com (Z.G.); liudi@haas.cn or liudi1963@163.com (D.L.); Tel./Fax: +86-451-87502330 (Z.G.)

[†]  These authors contributed equally to this work.

**Abstract:** Cold stress significantly affects gene expression in adipocytes; studying this phenomenon can help reveal the pathogeneses of conditions such as obesity and insulin resistance. Adipocyte triglyceride lipase (*ATGL*); cell death-inducing deoxyribonucleic acid (DNA) fragmentation factor subunit alpha (DFFA)-like effector (*CIDEA*); and uncoupling protein genes *UCP1*, *UCP2*, and *UCP3* are the most studied genes in pig adipose tissues under cold stress. However, contradictory results have been observed in gene expression changes to *UCP3* and *UCP2* when adipose tissues under cold stress were examined. Therefore, we conducted a meta-analysis of 32 publications in total on the effect of cold stress on the expression of *ATGL*, *CIDEA*, *UCP2*, and *UCP3*. Our results showed that cold stress affected the expression of swine adipocyte genes; specifically, it was positively correlated with the expression of *UCP3* in swine adipocytes. Conversely, expression of *ATGL* was negatively affected under cold stress conditions. In addition, the loss of functional *UCP1* in pigs likely triggered a compensatory increase in *UCP3* activity. We also simulated the docking results of UCP2 and UCP3. Our results showed that UCP2 could strongly bind to adenosine triphosphate (ATP), meaning that *UCP3* played a more significant role in pig adipocytes.

**Keywords:** beige adipose tissue; fat; pig; *UCP1*

## 1. Introduction

Human metabolic syndrome affects individuals with a high body weight (BW) [1]. The answer to the question of why individuals with a high BW are affected by human metabolic syndrome lies in the study of fat deposition mechanisms. Humans have two main types of fat tissues: white adipose tissue (WAT) and brown adipose tissue (BAT). WAT is primarily involved in energy storage, whereas BAT plays a role in heat production and the regulation of energy metabolism. BAT contains a large number of mitochondria [2]. Research on fat tissues has recently increased, and the conversion of WAT to BAT in particular has attracted widespread attention [3]. BAT plays an essential role in maintaining whole-body energy homeostasis by regulating insulin sensitivity and glucose metabolism [2]. We are interested in investigating how pigs with high-fat ratios maintain whole-body energy homeostasis. Pigs are important animal models for human diseases [4].

Cold stress significantly affects gene expression in adipocytes [5]. Expression of cold stress-related genes regulates the growth, differentiation, and function of adipocytes, thereby affecting the balance of energy metabolism in the body. In addition, WAT browning has demonstrated therapeutic potential for the treatment of human diseases [5]. Further study of the influence of cold stress on adipocyte gene expression can help reveal the pathogeneses of obesity, insulin resistance, and other ailments [6], which will provide

new insights into the treatment and prevention of these conditions [7]. Short-term cold stimulation leads to cold stress, whereas long-term cold stimulation results in adaptation to the cold. However, the mechanism by which WAT is converted to BAT and the duration of this conversion remain unclear. In this study, we did not distinguish between cold stress and adaptation to cold; this article uses the term "cold stress" consistently for both.

Pigs exposed to cold stress experience significant changes in adipose tissue gene expression. The most studied genes in adipose tissue gene expression under cold stress include adipocyte triglyceride lipase (*ATGL*); cell death-inducing deoxyribonucleic acid (DNA) fragmentation factor subunit alpha (DFFA)-like effector (*CIDEA*); and uncoupling protein genes *UCP1*, *UCP2*, and *UCP3*. Pigs have a defect in *UCP1* that prevents it from synthesizing protein [8]; however, that is not the focus of this study. *UCP2* and *UCP3* gene–encoded proteins are mainly located in the mitochondria and participate in energy metabolism [9]. They regulate oxidative phosphorylation (OXPHOS) and uncoupling, thereby affecting energy balance within cells. *ATGL* encodes an enzyme responsible for breaking down triglycerides in adipocytes, producing free fatty acids (FFAs) and glycerol [10]. Lipid metabolic products can also affect the expression of other genes. Adenosine triphosphate (ATP) regulates ATGL activity through the protein kinase A (PKA) pathway and inhibits or releases FFAs [11,12]; it also inhibits UCP2 and UCP3 activity in the mitochondria [13]. *CIDEA* encodes an enzyme involved in fatty acid ester degradation, which further regulates fat metabolism [14]. CIDEA helps regulate the fusion of lipid droplets, potentially regulating the process of adipocyte volume reduction [15].

It is reported that under cold stress, adipose-tissue *UCP2* decreases and *UCP3* increases [16]. However, Zhou et al. found that under cold stress, adipose-tissue *UCP3* decreased and *UCP2* increased [17]. These conflicting results led us to investigate, through a meta-analysis, the effect of cold stress on the expression of the swine adipocyte genes *ATGL*, *CIDEA*, *UCP2*, and *UCP3*.

## 2. Materials and Methods

### 2.1. Database Search Strategy and Study Inclusion

The articles included in this study were obligated to prioritize animal welfare by minimizing the uncomfortable reactions cold stimulation can cause in pigs. Two independent authors searched separately for the following keywords: "(pig OR, porcine OR swine OR hog OR boar OR sow OR piglet) AND (ATGL OR UCP2 OR UCP3 OR CIDEA) AND cold". We searched four databases—PubMed, ProQuest, ScienceDirect, and Web of Science—for literature published up to 1 October 2023. Table 1 presents the criteria for literature inclusion and exclusion, while Figure 1A illustrates the search and literature screening steps.

**Table 1.** Inclusion and exclusion criteria.

| Inclusion | Exclusion |
|---|---|
| Species evaluated included, but were not limited to, pigs | Pigs not used |
| English literature | Non-English |
| Analysis of *ATGL*, *CIDEA*, *UCP2,* and *UCP3* alone or with other genes in pigs | No melatonin treatment for pigs |
| Adipose tissue-data included | No adipose-tissue data |

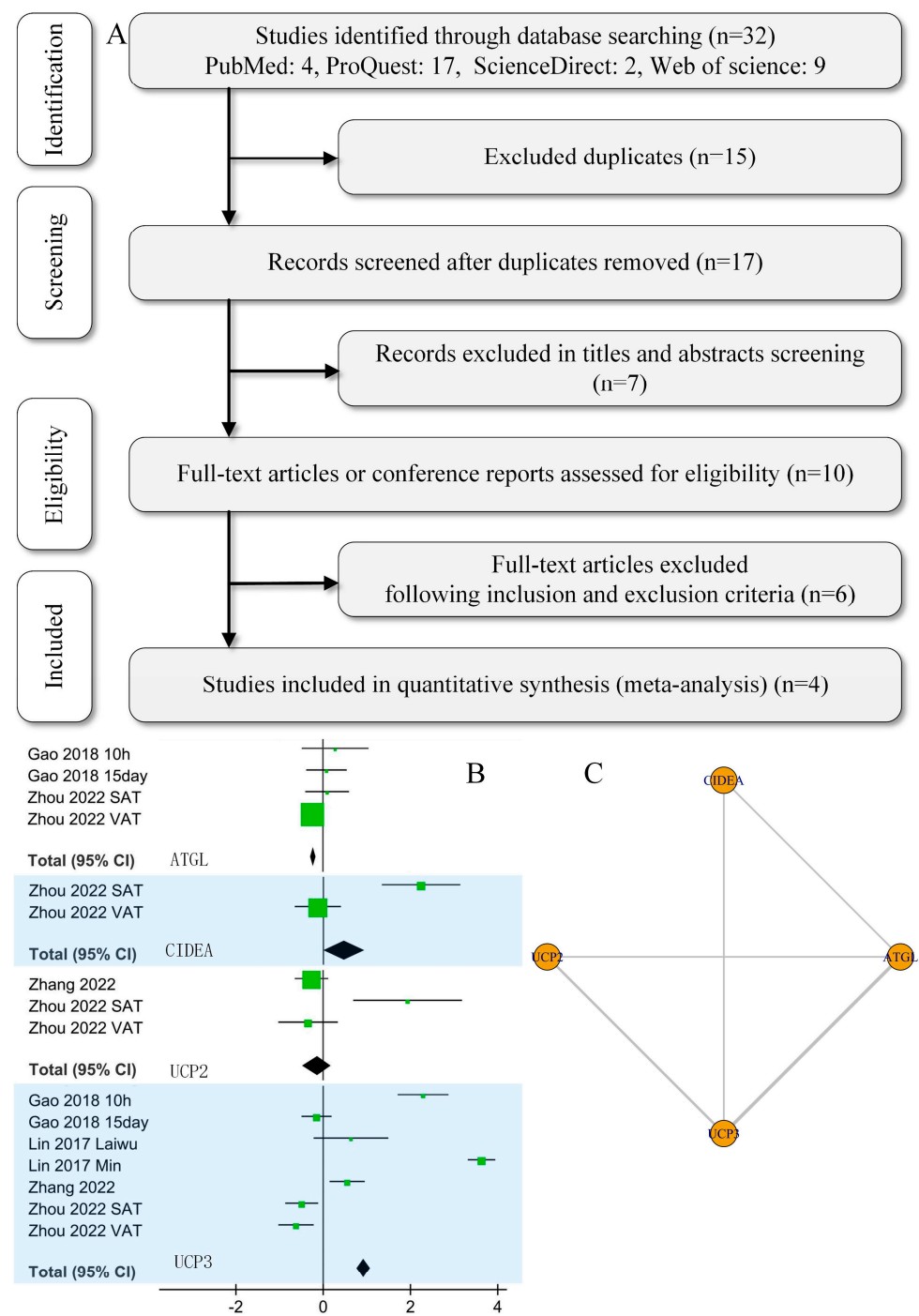

**Figure 1.** Summary of the study selection process and meta-analysis. (**A**) Literature selection process: Our search identified 32 publications based on keywords; ultimately, we used four articles. (**B**) Traditional meta-analysis of the effect of cold stress on *ATGL*, *CIDEA*, *UCP2*, and *UCP3* gene expression (Gao 2018 [18], Zhou 2022 [17], Lin 2017 [19], Zhang 2022 [16]). SAT: subcutaneous adipose tissue. VAT: visceral adipose tissue. The values of the invalid lines in the graph are 0, so when the 95% confidence interval (CI) was entirely greater than 0, cold stress was considered positively correlated with swine adipocyte gene expression. When the 95% CI was entirely less than 0, cold stress was considered negatively correlated with swine adipocyte gene expression. Green squares represent the weight of the study, and black diamonds represent the 95% CI. (**C**) Network diagrams comparing different genes. The thicker the line, the greater the number of studies. In this study, four datasets mentioned *ATGL* and *UCP3* simultaneously, and three mentioned *UCP2* and *UCP3* simultaneously.

### 2.2. Data Extraction

We extracted expression levels of pig *ATGL*, *CIDEA*, *UCP2*, and *UCP3* affected by cold stress from figures in the reviewed studies. GetData Graph Digitizer software (version 2.26) was used to extract the data. For example, we extracted Wuzhishan pig *UCP3* expression data (control, 0.99; standard deviation [SD], 0.4) from the Lin 2017 report [19]. The actual value of the control group was set at 1, and we carefully kept our error within a 95% confidence interval (CI). The extracted data included the number of experimental pigs, gene expression levels, and either SD or standard error (SE). If SE was obtained, it was converted to SD. Table 2 shows the characteristics of the included studies. We analyzed each treatment as an independent dataset [20]; for example, 10 h and 15 days in the Gao 2018 study were analyzed as two separate datasets [18]. The articles present the results of relative messenger ribonucleic acid (mRNA) expression, transcript expression levels, and transcripts per million (tpm) data, which were extracted by artificial means by two authors of the current study.

**Table 2.** Characteristics of the studies included.

|   | Study Year | Breed (Number) | Tissue | Temperature, Treatment Time | Genes | Age or Weight | Measurement |
|---|---|---|---|---|---|---|---|
| 1 | Gao 2018 [18] | NM (9) | Adipocytes | 4 °C, 10 h; 8 °C, 15 days | *UCP3*, *CIDEA*, *ATGL* | 5 days | Relative mRNA expression |
| 2 | Lin 2017 [19] | Wuzhishan (4), Min (4) | Adipocytes | 4 °C, 4 h | *UCP3* | 5 weeks | Relative mRNA expression |
| 3 | Zhang 2022 [16] | NM (5) | Fat tissue | 18 °C, 48 h | *UCP2, UCP3* | NM | Transcript expression levels |
| 4 | Zhou 2022 [17] | Duroc × Landrace × Yorkshire (6) | SAT, VAT | 5–7 °C, 14 h | *UCP2, UCP3, ATGL* | 120–125 kg | tpm |

NM: not mentioned; SAT: subcutaneous adipose tissue; VAT: visceral adipose tissue.

### 2.3. Traditional and Network Meta-Analyses

The effects of cold stress on the expression of *ATGL*, *CIDEA*, *UCP2*, and *UCP3* were determined through a traditional meta-analysis. We used Review Manager (Cochrane Collaboration v5.4; https://cochrane.org/ accessed on 21 September 2020) to calculate the total effect size of gene expression. Continuous data were used to determine the level of gene expression, and the 95% CI was calculated using the mean difference (MD).

We compared the expression of *ATGL*, *CIDEA*, *UCP2*, and *UCP3* under cold stress using a network meta-analysis, which was performed using the packages "coda", "rjag", and "gemtc" in R software (R Foundation for Statistical Computing, Vienna, Austria v4.3.2 31 October 2023). Gene expression data were normalized to fold change (FC) as follows (using *UCP3* as an example):

$$UCP3 \text{ gene expression} = \frac{\text{cold stress group } UCP3 \text{ expression}}{\text{control group } UCP3 \text{ expression}} \times 100\%$$

The corresponding 95% CIs were used to calculate effect sizes for continuous gene expression data.

### 2.4. Molecular Docking of UCP1, UCP2, and UCP3

We obtained human *UCP1* (NP_068605.1) and *UCP3* (NP_003347.1) amino acid sequences from the US National Center for Biotechnology Information (NCBI; Bethesda, MD, USA). SWISS-MODEL data (https://swissmodel.expasy.org/) were used to generate tertiary-protein structures. We downloaded ATP ligand structures from PubChem and performed docking using AutoDock Vina software v1.5.6 (https://vina.scripps.edu/ accessed

on 17 September 2014), with 100 iterations for more-reliable results. Amino acid sequences of pig *UCP2* (NP_999454.1) and *UCP3* (NP_999214.1) were downloaded from the NCBI, and the same method described above was used to perform comparative analysis.

## 3. Results

### 3.1. Traditional Meta-Analysis

The four studies retained for our meta-analysis include seven treatments [16–19]. Details are provided in Table 2. Due to one meta-analysis report including only one study [21], we decided to proceed with our analysis.

As shown in Figure 1B, the data indicated that cold stress was positively correlated with swine adipocyte *UCP3* gene expression (standard MD [SMD], 0.93; 95% CI, 0.78–1.08; $p < 0.001$). However, for *ATGL*, the correlation was negative (SMD, $-0.22$; 95% CI, $-0.29$ to $-0.15$; $p = 0.17$). Cold stress did not correlate at all with *CIDEA* (SMD, 0.46; 95% CI, $-0.01$ to 0.94; $p = 0.04$) or *UCP2* (SMD, $-0.14$; 95% CI, $-0.46$ to 0.19; $p = 0.41$).

### 3.2. Network Meta-Analysis

The network diagram in Figure 1C compares gene expression among the swine adipocyte genes *ATGL*, *CIDEA*, *UCP2*, and *UCP3*. Trace plots with 25,000 iterations are shown in Figure 2A. Markov Chain Monte Carlo (MCMC) convergence was excellent. The normal distribution of density plots with 5000 iterations did not display a double-peak phenomenon (Figure 2B). Furthermore, the Brooks–Gelman–Rubin (BGR) diagnosis plot (Figure 2C) shows that the potential scale reduction factor was between 1.00 and 1.01 after 22,000 iterations, indicating that the running outcomes of the network meta-analysis were reliable. Figure 3 shows that cold stress had a consistent effect on the expression of *ATGL*, *CIDEA*, *UCP2*, and *UCP3* without any order of priority.

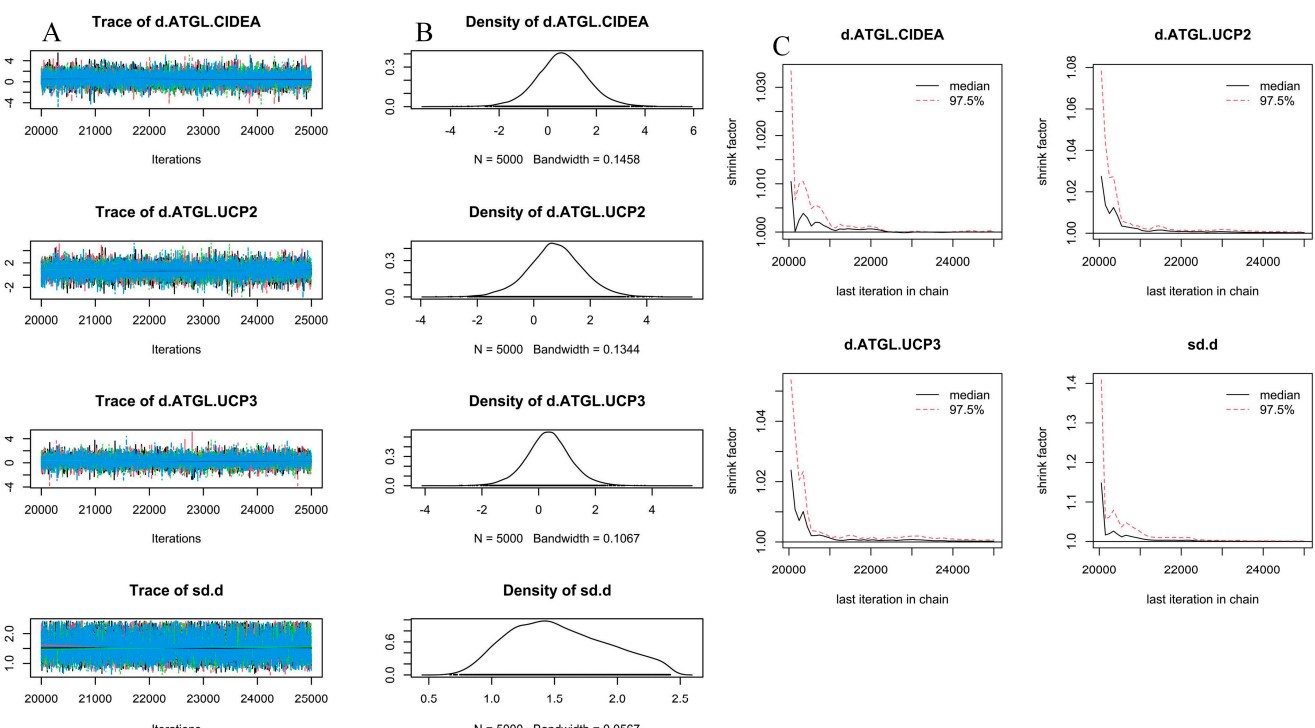

**Figure 2.** Network meta-analysis credibility. (**A**) Trace plots of network meta-analysis: Overlapping areas are numerous in MCMC, making it impossible to identify individual chains. (**B**) Density plots of network meta-analysis. With 5000 iterations, each result showed a single peak. We did not observe the double-peak phenomenon, indicating a high level of confidence in the computational results. (**C**) BGR diagnosis plot. After 5000 iterations, the shrink factor of each result quickly converged to 1.00 and remained stable, indicating that the model had achieved the expected level of convergence.

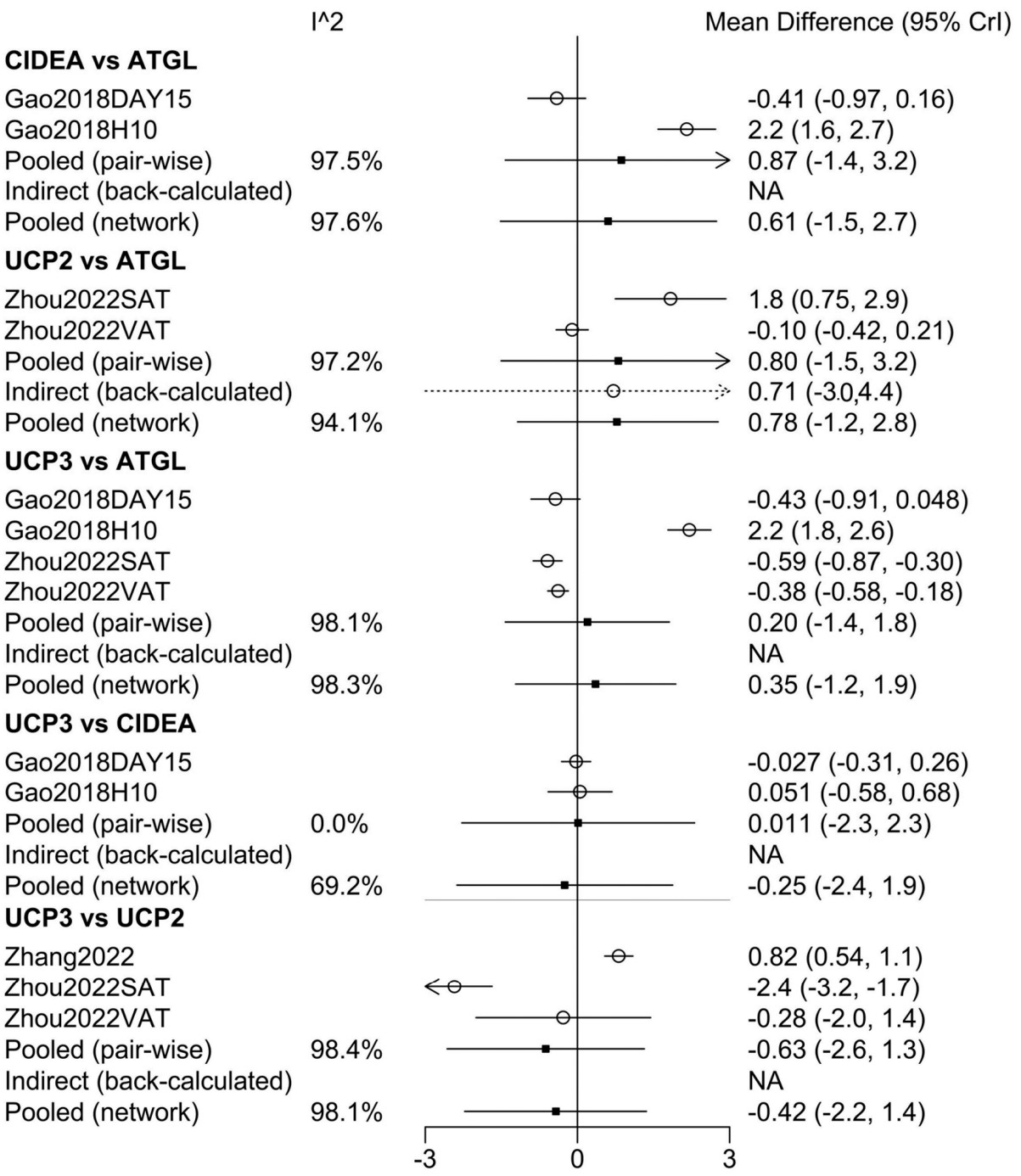

**Figure 3.** Forest plot of network meta-analysis. Study results showed no priority ranking among *ATGL*, *CIDEA*, *UCP2*, and *UCP3* in the effect of cold stress on swine adipocyte gene expression; the results were consistent. Specifically, results for CIDEA vs. ATGL (SMD, 0.61; 95% CI, −1.5 to 2.7), UCP2 vs. ATGL (SMD, 0.78; 95% CI, −1.2 to 2.8), UCP3 vs. ATGL (SMD, 0.35; 95% CI, −1.2 to 1.9), UCP3 vs. CIDEA (SMD, −0.25; 95% CI, −2.4 to 1.9), and UCP3 vs. UCP2 (SMD, 0.42; 95% CI, −2.2 to 1.4) all reached the null value of 0. (Gao 2018 [18], Zhou 2022 [17], Zhang 2022 [16]).

### 3.3. Molecular Docking of UCP1, UCP2, and UCP3

Comparative results of human UCP1 and UCP3 protein docking are shown in Figure 4. UCP3 (Figure 4B) bound to ATP more tightly than UCP1 (Figure 4A), while porcine UCP2 (Figure 4D) bound to ATP more strongly than UCP3 (Figure 4C). UCP1 was more active than UCP3 in human cells. We believe this would result in increased UCP3 activity in pig adipocytes.

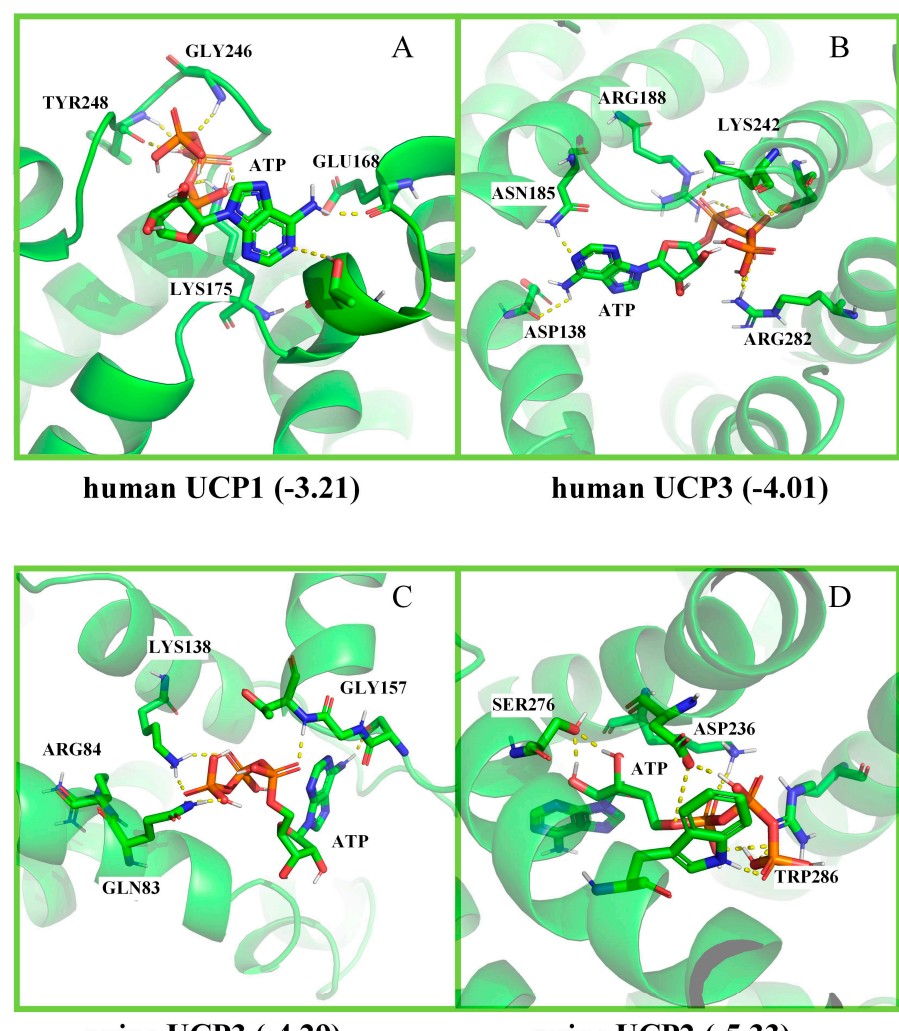

**Figure 4.** Molecular docking of UCP1, UCP2, and UCP3. The binding affinities of human UCP1 and UCP3 were −3.21 and −4.01 kcal/mol, respectively, while those of swine UCP3 and UCP2 were −4.29 and −5.33 kcal/mol, respectively. In swine, the number of amino acids involved in the binding pocket of molecular docking exceeded that in humans, and the degree of binding was also tighter than in humans. The dashed line in the figure represents a connection bond of <5 Å.

## 4. Discussion

Cold stress–induced BAT activation is believed to improve human metabolic health [22]. Cold stress increases levels of nicotinamide adenine dinucleotide ($NAD^+$); this can further promote mitochondrial heat production [22,23], which can enhance immunity.

### 4.1. Effect of Cold Stress on Adipocyte Gene Expression

Cold stress signals affect adipocyte gene expression by activating pathways in the nervous and endocrine systems. Research has found that cold stress can activate the hypothalamic–pituitary–adrenal (HPA) axis, promoting corticosterone secretion [24]. Corticosterone can regulate adipocyte gene expression, thereby affecting adipocyte growth, differentiation, and function [25]. Corticosterone can increase the expression of *ATGL* in broiler chicken WATs [26]. However, our results showed a negative correlation between cold stress and *ATGL*, indicating that the cold stress mechanism in broiler chicks is different from that in pigs.

Expression of certain adipocyte genes changes under cold stress conditions [5,16]. These genes are crucial in regulating the growth, differentiation, and function of adipocytes under cold stress. For example, heat shock factor *1* (*HSF1*) protects cell membrane stability,

reduces intracellular oxidative stress (OS) levels, and alleviates cold stress damage to adipocytes [27]. Changes in *UCP2* expression are generally believed to be caused by cold stress. However, our results showed no relationship between cold stress and *CIDEA* and *UCP2* expression. Additional research might be required to validate this finding.

Although cold stress promotes pig *UCP1* gene expression [18], the pig *UCP1* gene is defective and cannot synthesize protein [8], meaning that it can be detected only at the gene expression level. Similar studies have shown that *protein kinase D1* (*Prkd1*) deletion in mouse BAT maintains thermogenesis after cold exposure [28], suggesting compensatory pathways in adipose tissue. Our results showed a positive correlation between cold stress and swine adipocyte *UCP3* gene expression, indicating that the high activity of *UCP3* might be due to the loss of function of *UCP1* in pigs.

### 4.2. Transformation of Pig Adipocytes and Induction Factors

BAT contains more mitochondria than WAT [2]. Genes expressed in pig WAT include those involved in BAT differentiation in the natural growth state [29]. While there are no reports on pigs with BAT [30], pigs might possess beige (brite adipose tissue) [19], which could have a mutual-conversion relationship with WAT [5]. Our published articles indicate that cold stress can promote WAT browning [18]. In addition, rosiglitazone and rapamycin can induce WAT in beige adipocytes in vitro [31]. Rosiglitazone, T3, and indomethacin can reportedly also induce beige adipocytes in mouse WAT [32].

### 4.3. Mutual Regulation of ATGL, CIDEA, UCP2, and UCP3

The proposed mechanisms of action of *ATGL*, *CIDEA*, *UCP2*, and *UCP3* in swine adipocytes are shown in Figure 5.

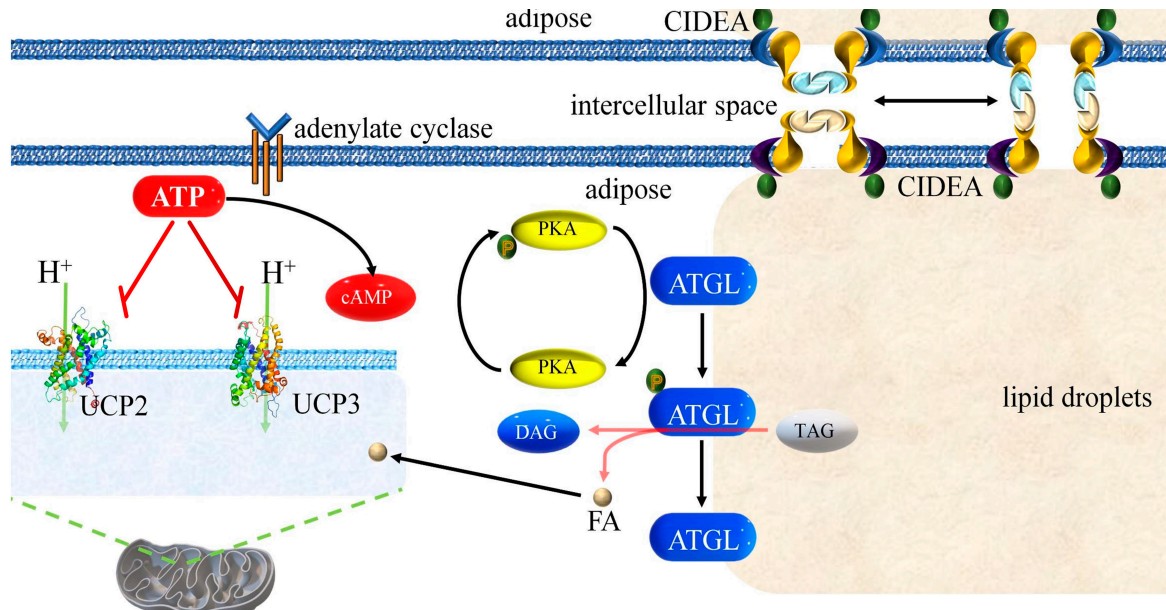

**Figure 5.** Proposed mechanisms of action of ATGL, CIDEA, UCP2, and UCP3 in swine adipocytes. Lipid droplet transfer between multiple adipocytes was accomplished using CIDEA; this transfer is mediated by the closure or opening of the CIDEA channel between adjacent adipose cells. PKA promotes phosphorylation of ATGL and, after the release of ATGL's phosphate, mediates the breakdown of TGs into DAGs and FFAs. *UCP2* and *UCP3*, located on the surface of mitochondria, pump hydrogen ions into the mitochondria and simultaneously release heat. FFA: free fatty acid. PKA: protein kinase A. TG: triglyceride. DAG: diacylglycerol. The red dashed line indicates inhibition.

*ATGL*, *CIDEA*, *UCP2*, and *UCP3* participate in regulating fat metabolism and energy balance. *ATGL* and *CIDEA* jointly regulate fat metabolism, produce metabolic products, and

affect *UCP2* and *UCP3* expression. The main function of *UCP2* and *UCP3* in mitochondria is to produce heat [9]. These genes work together to regulate fat metabolism and energy balance, which helps maintain homeostasis in organisms. Our network meta-analysis results showed no strong causal relationship among these four genes under cold stress conditions.

The results of our molecular-docking experiment indicated that in pig adipocytes, UCP2 is bound to ATP more strongly than UCP3. Similarly, in humans, UCP3 is bound to ATP more tightly than UCP1. This difference in ATP binding capacity between UCP1 and UCP3 in human adipocytes can reportedly result in UCP3 playing a role only when UCP1 cannot fulfill its physiological function [33]. Therefore, we hypothesized that pig adipocytes have high UCP3 activity and that UCP2 plays a role only when UCP3 cannot fulfill its physiological function. If this is the case, then in pig adipocytes, *UCP2* plays a more significant role than *UCP3*.

In summary, swine adipocyte *UCP3* gene expression was found to be positively correlated with cold stress, while *ATGL* expression was negatively correlated with cold stress. We found no correlation between cold stress and the expression of *UCP2* and *CIDEA* in swine adipocytes. During the early stages of cold stress, we hypothesized that changes in *UCP2* and *CIDEA* expression that occur might cause this phenomenon. Then, adipocyte volume decreased with time. *CIDEA* was no longer needed to regulate the fusion of lipid droplets, and gene expression returned to normal levels. Similarly, expression of *UCP2*, which initially functioned together with *UCP3*, also returned to normal levels because *UCP3* could independently fulfill its physiological function.

## 5. Limitation

The study used four published articles that employed relative gene mRNA expression, transcript expression levels, and transcripts per million data. The data underwent normalization during the network meta-analysis, which might have led to errors. The aim of the network meta-analysis was to identify the genes most affected by cold stress and then rank them. We saw no clear differences among *ATGL*, *CIDEA*, *UCP2*, and *UCP3*, which is a result we can accept. *UCP3* is reported to have a higher expression level in skeletal muscle [34] and the liver. However, expression of *UCP3* in skeletal muscle under cold stress was considered in the meta-analysis [16].

## 6. Conclusions

The gene expression of swine adipocyte *UCP3* was positively affected, and that of ATGL was negatively affected, by cold stress. Network meta-analysis results showed that cold stress affected the expression of *ATGL*, *CIDEA*, *UCP2*, and *UCP3*. One included study shows that cold stress promotes expression of the pig *UCP1* gene [18], but this gene cannot synthesize protein, and its failure to do so might lead to high UCP3 activity as a compensatory mechanism.

**Author Contributions:** L.L. and Z.G. collected the data and conducted the analysis. Z.G. and D.L. conceived this research. H.M. created the figures. L.W., B.F., and F.W. reviewed the draft. All authors have read and agreed to the published version of the manuscript.

**Funding:** This study was supported by the National Natural Science Foundation of China (No. U20A2052), the National Center of Technology Innovation for Swine (No. NCTIP-XD1C16), and the Heilongjiang Provincial Scientific Research Business Fund Project (No. CZKYF2023-1-C004). These funding agencies were not involved in the development of the study design or the preparation of this manuscript.

**Institutional Review Board Statement:** Not applicable.

**Informed Consent Statement:** Not applicable.

**Data Availability Statement:** Please contact the author Zhenhua Guo to request the data.

**Conflicts of Interest:** The authors declare that they have no conflicts of interest to report.

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
