# Peer review of "Network Meta-Analysis: Effect of Cold Stress on the Gene Expression of Swine Adipocytes ATGL, CIDEA, UCP2, and UCP3"

_cimb, doi:10.3390/cimb46050240_

Round 1

Reviewer 1 Report

Comments and Suggestions for Authors

1. Is it reasonable that only four articles were included for the meta-analysis?

2. How was the expression of cold stress marker genes (ATGL, CIDEA, UCP2, and UCP3) calculated? Was the data extracted from published figures? If so, authors might have obtained only the expression trends of these genes. What's the significance of this approach?

3. In line 109, "The gene expression data, such as UCP3, were normalized as follows." I believe it's not a data normalization method but rather for calculating the fold change of UCP3 expression.

4. In line 124, for the correlated analysis, how many samples were used, and what parameters were considered? How was the bias caused by different cold treatments (e.g., acute cold exposure, long-time cold exposure) addressed?

5. In the abstract and graphical abstract, "a more significant role for UCP3" was mentioned, but it wasn't elaborated later. Can you discuss the role of UCP3 in adipose tissue response to cold stress?

6. UCP3 is reported to have a higher expression level in skeletal muscle. Was the expression of UCP2 and UCP3 in skeletal muscle under cold stress considered in the analysis?

7. In the discussion (Section 4.2), the transformation of pig adipocytes was discussed. Can you explain the relationship between the genes involved in this manuscript and white adipose tissue (WAT) or beige adipocytes?

Comments on the Quality of English Language

English language improvements are needed. For instance, on page 1, line 21, and page 2, line 68, "adipocyte tissues" should be "adipose tissues." Also, on page 1, lines 30-31, change "We analyzed ATGL, CIDEA, UCP2, and UCP3 to determine the effect of cold stress on the expression of swine adipocytes." to "We analyzed the effect of cold stress on the expression of ATGL, CIDEA, UCP2, and UCP3 genes in swine adipocytes."

Author Response

Revised manuscript cimb-2947384 " Network meta-analysis: Effect of cold stress on swine adipocytes ATGL, CIDEA, UCP2, and UCP3 gene expression"

Dear Ms.Worawalan Bunyamalik Editor and reviewers:

On behalf of my co-authors, thank you for giving us the opportunity to revise our manuscript. We appreciate the editor and the reviewers’ positive and constructive comments and suggestions for our manuscript.

The manuscript has already been corrected and is attached for further evaluation.

We would like to express our great appreciation to you and the reviewers for your comments on our paper. We look forward to hearing from you.

Thank you and best regards.

Yours sincerely,

Guo

Animal Husbandry Research Institute of Heilongjiang Academy of Agricultural Sciences

368 Xuefu Road, Harbin, P.R.China, 150086

Office Tel: 086-451-87502330

Mobile:  086-13115607125

Reviewer 2 Report

Comments and Suggestions for Authors

The study investigated the effect of cold stress on several genes expression in swine adipocyte, including ATGL, CIDEA, UCP2, and UCP3.  The authors performed Network meta-analysis on the data obtained from 4 literatures, from which they found that cold stress could affect UCP3 and ATGL expression differently. The comments are below:

 Major:

1.      Why the UCP protein activity can be affected by the tightness of the binding with ATP. Please provide some evidence to support the hypothesis.

2.      On the conclusion part, the author described that “Our results show that cold stress promotes the expression of the pig UCP1 gene”. However, there was no any pig UPC1-related data I can find in the manuscript. Please provide some related data.

3.      Please provide much more detail information in each figure legend. Also, there lacked the comprehensive description about Fig 5.

Minor:

1.      The different roles of ATGL, CIDEA, UCP2 and UCP3 in adipocytes should be demonstrated in introduction part instead of the discussion part.

2.      It’s better to explain the experimental methods in as much detail as possible, such as the “artificial means in data extraction”.

Comments on the Quality of English Language

1.There was a grammar mistake in line 114. 

2.There should be negative instead of positive in line 126 about the ATGL.

Author Response

(The authors gave the same response as above.)

Reviewer 3 Report

Comments and Suggestions for Authors

In their study, Zhenhua Guo and colleagues aimed to conduct a meta-analysis on the impact of cold stress on the expression of ATGL, CIDEA, UCP2, and UCP3 genes. They found that cold stress positively influenced the expression of the UCP3 gene in swine adipocytes and negatively influenced ATGL expression. The network meta-analysis indicated that cold stress modulates the expression of these genes. Additionally, their findings suggest that cold stress enhances the expression of the UCP1 gene in pigs. However, several issues within the paper need to be addressed:

1. Clarification of paper type: Although labeled as a review, the structure and results suggest the paper is more akin to a research article.

2. Selection rationale for genes: The paper should provide more comprehensive background information to justify the focus on only ATGL, CIDEA, UCP2, and UCP3, given the multitude of genes involved in porcine adipose tissue response to cold stress.

3. Figure 1 readability: It is recommended that Figure 1 be reorganized to enhance legibility, as the current font size is too small.

4. Placement of statements: The assertions made in lines 121-122 are misplaced and should be relocated to a more appropriate section within the paper.

5. Figure labeling: If the authors wish to discuss Figure 1B before Figure 1A, they should consider renumbering the figures accordingly for logical coherence.

6. Coverage of network meta-analysis: Despite the title suggesting a focus on network meta-analysis, only a brief mention is made in lines 136-138. This section should be expanded to reflect its purported significance.

7. Supporting evidence for conclusions: The conclusion drawn in line 250 regarding UCP1 failure potentially leading to increased UCP3 activity as a compensatory mechanism lacks supporting results. The paper should include relevant data or references to substantiate this claim.

Comments on the Quality of English Language

minor editing of English is required

Author Response

(The authors gave the same response as above.)

Reviewer 4 Report

Comments and Suggestions for Authors

In the present manuscript, Guo et al. analyzed expression data from genes coding for four proteins involved in the cold stress response in pig adipocytes. Four published studies were utilized. ATP binding to predicted structures of these proteins was simulated. From gene expression analysis and binding simulations, a functional interplay mechanism was proposed for these four proteins.

The present study addresses the impact of these proteins in cold stress in the framework of metabolic syndrome, which is a pertinent field of study. Nevertheless, the conclusions are not supported by any new experimental data (only gene expression analysis and computational simulations are provided), which should be acknowldged as a limitation of the study. The literature selection and methods are clearly described. However, results need to be described in more detail, as explained in some of the remarks below.

Specific remarks:

1. In the abstract, please include the full names of ATGL, CIDEA, UCP1, UCP2, and UCP3.

2. In line 90, please consider correcting "We extracted the expression of pig ATGL, CIDEA, UCP2, and UCP3 genes (...)" to "We extracted the expression levels of pig ATGL, CIDEA, UCP2, and UCP3 genes (...)".

3. In line 92, in "(...) data from the figure", please include the figure number.

4. In Table 2, please inlcude the reference number (as listed in the references section) of each study.

5. In the legend of Figure 1B, please explain the meaning of the green squares and black diamonds.

6. In the horizontal axis of Figure 1B, please indicate what statistical parameter is represented. Please also make differences in line thickness more evident.

7. In the legends of Figures 1B and 1C, please describe these figures in more detail, since they may not be fully intuitive to the reader.

8. Please divide Figure 4 into panels and mention each one in the main text.

9. In lines 151-152, please better justify the statements "UCP3 binds ATP more tightly than UCP1" and "Porcine UCP2 binds to ATP more strongly than UCP3". Since these results are computational simulations rather than experimental determinations, it might be better to mention "is predicted to bind" intead of "binds". Please also mention what is meant by active in "UCP1 is more active than UCP3 in human cells".

10. In the legend of Figure 4, please describe each panel of the figure in greater detail, including the molecular interactions. Please also correct "5A" to "5 Å".

11. In the legend of Figure 5, please include all abbreviations used in this figure.

12. In lines 233-234, please include a reference for "However, the adipocyte volume decreases with time". If it is an hypothesis, please mention that.

13. In line 243, please correct "between" to "among", since the comparison includes more than two elements.

Comments on the Quality of English Language

(Please see above)

Author Response

(The authors gave the same response as above.)

Round 2

Reviewer 1 Report

Comments and Suggestions for Authors

1. What’s the meaning of “UCP2 bound to ATP, meaning that UCP3 played a more significant role in pig adipocytes.” This sentence makes no sense, how to conclude that?

2. “Our results showed that cold stress affected the expression of swine adipocyte genes; specifically, it was positively correlated with expression of UCP3 in swine adipocytes.” In this sentence, what parameter was used to calculate the correlated analysis with expression of UCP3? What kind of data were used for the correlated analysis between UCP3 expression and ? cold conditions? If used cold conditions, how to calibrate the long-term and short-term cold stimulation as well as the impact of different temperatures? Please rewrite this part in method.

Comments on the Quality of English Language

no

Author Response

Round 2----Revised manuscript cimb-2947384 " Network meta-analysis: Effect of cold stress on swine adipocytes ATGL, CIDEA, UCP2, and UCP3 gene expression"

Dear Ms.Worawalan Bunyamalik Editor and reviewers:

On behalf of my co-authors, thank you for giving us the opportunity to revise our manuscript. We appreciate the editor and the reviewers’ positive and constructive comments and suggestions for our manuscript.

The manuscript has already been corrected and is attached for further evaluation.

We would like to express our great appreciation to you and the reviewers for your comments on our paper. We look forward to hearing from you.

Thank you and best regards.

Yours sincerely,

Guo

Animal Husbandry Research Institute of Heilongjiang Academy of Agricultural Sciences

368 Xuefu Road, Harbin, P.R.China, 150086

Office Tel: 086-451-87502330

Mobile:  086-13115607125

Reviewer 2 Report

Comments and Suggestions for Authors

The author has addressed all of my questions.

Author Response

Round 2----Revised manuscript cimb-2947384 " Network meta-analysis: Effect of cold stress on swine adipocytes ATGL, CIDEA, UCP2, and UCP3 gene expression"

Dear Ms.Worawalan Bunyamalik Editor and reviewers:

On behalf of my co-authors, thank you for your work.

Yours sincerely,

Guo

Animal Husbandry Research Institute of Heilongjiang Academy of Agricultural Sciences

368 Xuefu Road, Harbin, P.R.China, 150086

Office Tel: 086-451-87502330

Mobile:  086-13115607125

Reviewer 3 Report

Comments and Suggestions for Authors

No more questions.

Author Response

(The authors gave the same response as above.)
